# Prolonged Survival and Restored Useful Life by Early Induction of Intrathecal Chemotherapy in a Patient with Leptomeningeal Carcinomatosis from Ovarian Cancer

**DOI:** 10.3390/brainsci12060748

**Published:** 2022-06-07

**Authors:** Kento Takahara, Makoto Katayama, Ryota Tamura

**Affiliations:** 1Department of Neurosurgery, Kawasaki Municipal Hospital, Kawasaki 210-0013, Japan; pami529@yahoo.co.jp (K.T.); makoto.katayama@gmail.com (M.K.); 2Department of Neurosurgery, Keio University School of Medicine, Tokyo 160-8582, Japan

**Keywords:** early diagnosis, intrathecal chemotherapy, leptomeningeal carcinomatosis, ovarian cancer

## Abstract

Leptomeningeal carcinomatosis (LMC) is a rare but devastating complication of advanced cancer. Breast cancer, lung cancer, and melanoma are the three most common causes of LMC, whereas it is rare in ovarian cancer. Here, we report the case of a 59-year-old woman who was diagnosed with LMC from ovarian cancer and was successfully treated with intrathecal chemotherapy via Ommaya reservoir and radiation therapy. The patient had an amelioration of symptoms and prolonged survival. Though LMC from ovarian cancer is thought to be rare, it is not going to remain a rare entity because the incidence of LMC in general is thought to be increasing, which is also the case with ovarian cancer. According to 31 cases whose treatment course is reported in literature, despite the absence of an established treatment for LMC, intrathecal (IT) chemotherapy whose survival benefit has been suggested in past studies might also prolong survival in patients with LMC from ovarian cancer. IT chemotherapy via Ommaya reservoir may be preferred to the lumbar puncture route. The presentation of non-specific symptoms of LMC in patients may hinder its diagnosis; however, early diagnosis and treatment induction is the key for patients’ prolonged survival and restored useful life.

## 1. Introduction

Leptomeningeal carcinomatosis (LMC) is a rare but devastating complication of advanced cancer. It is diagnosed in approximately five percent of patients with metastatic cancer [1,2,3]. Patients with LMC have a poor prognosis; the median survival after diagnosis is known to be a few months [4,5]. Malignant cells are disseminated via the cerebrospinal fluid (CSF) and spread throughout the subarachnoid space, causing multifocal signs and symptoms often misdiagnosed as a side effect of cancer-treating agents resulting in a delay in treatment initiation. Treatment for LMC includes intrathecal (IT) or high-dose chemotherapy, radiation therapy to areas of bulky disease or CSF flow obstruction, and control of increased intracranial pressure (ICP). However, the optimum therapy remains poorly defined because of the paucity of prospective randomized trials [6]. Breast cancer, lung cancer, and melanoma are the three most common causes of LMC, whereas it is rare in ovarian cancer [6]. Herein, we present a case of LMC from ovarian cancer successfully treated with the intrathecal administration of methotrexate via an Ommaya reservoir and radiation therapy.

## 2. Case Presentation

We report a case of a 54-year-old female, diagnosed with stage IV ovarian cancer and who underwent total hysterectomy, bilateral salpingo-oophorectomy, retroperitoneal lymphadenectomy, and omentectomy. She was administered paclitaxel and carboplatin pre- and post-chemotherapy. However, recurrent lesions were detected in her pelvis 2.5 years after the first surgery, for which debulking was performed again. Due to recurrence, the regimen was changed to gemcitabine, olaparib, and liposomal docetaxel. However, two years after the second surgery, at the age of 59, she visited a nearby doctor complaining about headache and nausea. Magnetic resonance imaging (MRI) revealed an enhanced lesion on the upper surface of her cerebellum spreading along the sulci. With the suspicion of CNS involvement of the ovarian cancer, she was referred to our department. Her Karnofsky performance status (KPS) score was at 80% during chemotherapy during which she developed headache and nausea and this prompted its stop. Her cognitive function then started to deteriorate rapidly, and she acquired gait disturbance in a few weeks with a KPS score of 40%. Given the MRI scan displaying a “diffuse classical” pattern [7] (Figure 1), as well as the symptomatology being suggestive of LMC, a CSF sample was obtained and analyzed, which revealed malignant cells, establishing the diagnosis of LMC. An Ommaya reservoir implantation was performed, and intrathecal methotrexate was administered 10 days after the onset of her headache and nausea. Her headache and nausea were relieved shortly after the induction of IT chemotherapy. Her cognitive function also gradually improved making her independent again (KPS 80%). Her CSF CA125 was rapidly decreased after induction of IT chemotherapy (Figure 2), paralleling with her clinical symptoms. Since her CSF CA125 remained low, a follow-up MRI revealed a slight enlargement of gadolinium-enhanced leptomeningeal lesions three months after the introduction of IT chemotherapy. The decision was to do a whole brain irradiation to control this bulky lesion. One month after the whole brain irradiation, the CSF cytology turned negative for the first time after LMC diagnosis, and at the same time CSF CA125 reached “normal level” (<4.3 U/mL) reported elsewhere [8,9]. Intrathecal chemotherapy was continued to maintain her status, until it was not possible to continue our outpatient clinic, mainly because of cancer-related pain from peritoneal dissemination nine months after the introduction of IT chemotherapy. She passed away 10 months after LMC diagnosis due to progressive disease in her abdominal cavity, with her CSF lesion left considerably dormant.

## 3. Discussion

Here, we described a case of LMC, an ovarian cancer complication which was successfully treated with IT chemotherapy via an Ommaya reservoir and with radiation therapy. Longer survival and symptom relief were achieved by early diagnosis and treatment induction.

LMC from ovarian cancer is a rare complication and therefore is infrequently recognized as a target for treatment; however, the situation may be changing. Previously, Yust et al. reported that only 0.06% (8/13126) of patients with ovarian cancer developed LMC [10]. However, Yano et al. reported that the incidence of LMC from gynecological cancers may be much higher than what is currently perceived. According to a report based on the brain tumor registry of Japan (2005–2008), lung and breast cancers were the most common sources of brain metastases [11]. The frequency of gynecological cancers among all cases of brain metastases was 4.6% which was similar to the incidence of gynecological cancers in Japan during the period 2005–2008 (3.9–4.1%). This implies that the propensity of gynecological cancers to metastasize to the CNS may be average rather than low. LMC may not have been previously suspected in gynecological cancers due to the belief of a rare frequency and the poor prognosis of patients receiving only palliative care in the terminal state. However, analogous to the CNS metastases from ovarian cancer, the incidence of LMC may be much higher than what is currently observed [12]. It has been repeatedly pointed out that the incidence of LMC from solid cancers such as lung and breast cancer is increasing; whereas, the continuous improvement of systemic treatment over the years utilizing multiple large molecular agents has managed to keep the disease dormant and rewarding the patient with a longer survival. However, the poor penetration of these molecules through the blood–brain barrier (BBB) leave the disease in the CNS, which eventually accumulates and causes LMC [3,6,13,14,15,16]. The systemic treatments currently used to treat gynecological cancers which include cytotoxic agents such as platinums and taxanes [17,18,19,20,21,22] are similar and are agents that are likely to leave the disease in this sanctuary site. The more effective the regimen gets, the more likely this phenomenon will be encountered. These facts may suggest that LMC from ovarian cancer is not going to remain a rare entity.

In our case, IT chemotherapy via an Ommaya route prolonged survival. No established treatment exists for LMC; however, several retrospective studies involving breast cancer and lung cancer have demonstrated a survival benefit using IT chemotherapy [23,24]. Currently, 31 cases of LMC from ovarian cancer in which the treatment option is available have been reported in the literature (Table 1) [25,26,27,28,29,30,31,32,33,34,35,36,37,38,39,40,41,42,43,44,45,46,47,48,49,50,51,52]. According to the case reports summarized in Table 1, the median survival after LMC diagnosis is more than seven months in patients with IT chemotherapy and three weeks in patients without, possibly suggesting the prolonged survival by IT chemotherapy. In addition, among the patients treated with IT chemotherapy, the median survival after LMC diagnosis is 10 months in cases with an Ommaya reservoir, while it is 3.5 months in cases treated via lumber puncture route. This suggests that a prolonged survival is achieved by IT chemotherapy via an Ommaya reservoir. This idea is supported by the recent retrospective study of LMC from various types of primary cancers showing significant superiority of IT chemotherapy via Ommaya reservoir in overall survival to IT chemotherapy via lumbar puncture (9.2 vs. 4.2 months) [53]. As can be observed from the case report, it is likely that IT chemotherapy via an Ommaya reservoir is the preferred treatment for LMC. These cases with IT chemotherapy reported in the literature are likely to be successful cases and with relatively preserved performance status at diagnosis. However, we would like to emphasize the feasibility and potential of IT chemotherapy via Ommaya reservoir for restoring independent and useful life in LMC patient, who mostly live miserable remaining life without treatment for LMC even with preserved PS at diagnosis.

In this case, radiation therapy was also performed for the bulky lesions. Radiation is geared towards symptom management due to bulky lesions typically in the spine, facilitating the use of IT chemotherapy by restoring CSF flow and reducing tumor bulk [24,54], though the radiation itself does not prolong survival [6].

In our case, early diagnosis with MRI and CSF study prior to rapid deterioration of her performance status might have a positive effect on the responsiveness to IT chemotherapy. LMC can be detected using MRI with a sensitivity of 70% [55,56,57], and by CSF cytology with a sensitivity of >90% if it is properly performed [58,59]. The CSF CA125 was also useful for diagnosis and paralleled with her clinical course (Figure 2), suggesting the potential of CSF CA125 as a surrogate marker of leptomeningeal disease. Despite these credible diagnostic tools, the presentation of LMC, which includes minor neurological symptoms may hinder the clinician’s decision of performing these diagnostic tools for LMC, often resulting in the delay of its diagnosis. It is known that the death rate increases by 19% per 10-unit decrease in KPS, and therefore, a delay in treatment induction which will result in further neurological deterioration is critical [60,61]. Yano et al. emphasized the importance of the early diagnosis of LMC from gynecological cancer, considering the importance of preserved performance status at diagnosis [12]. We managed to detect the disease in this case, which resulted in a significant improvement in the patient’s performance status, allowing a period of useful life. Clinicians should not hesitate to refer to neurologists if they encounter possible symptoms.

## 4. Conclusions

In this report, we would like to notify clinicians that LMC from ovarian cancer may no longer be rare, and we would like to highlight the importance of early diagnosis of LMC from ovarian cancer by showing that, an improved outcome is achieved with aggressive treatment. In treating LMC from ovarian cancer, IT chemotherapy via Ommaya reservoir may be feasible and effective.

## Figures and Tables

**Figure 1 brainsci-12-00748-f001:**
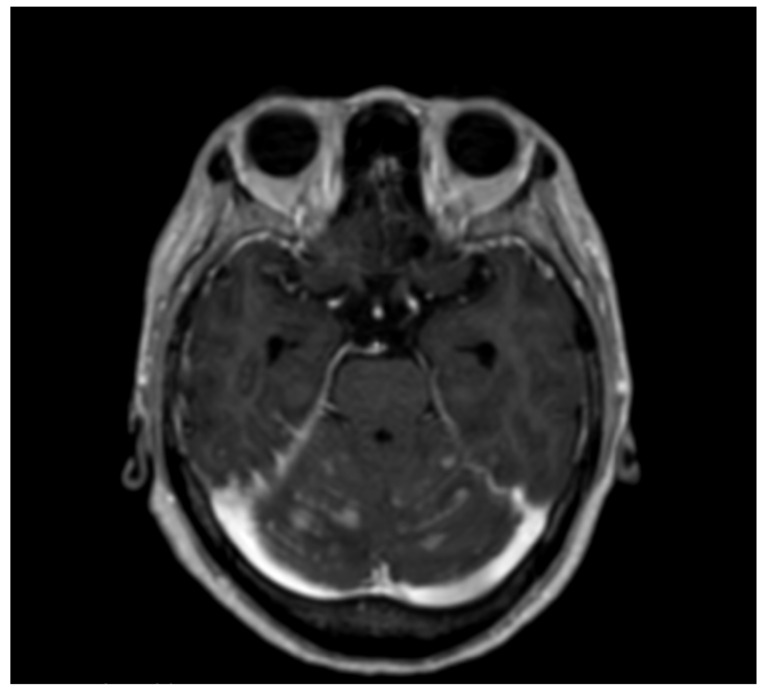
T1 weighted image with Gadolinium enhancement. Leptomeningeal enhancement along the sulci in the cerebellum.

**Figure 2 brainsci-12-00748-f002:**
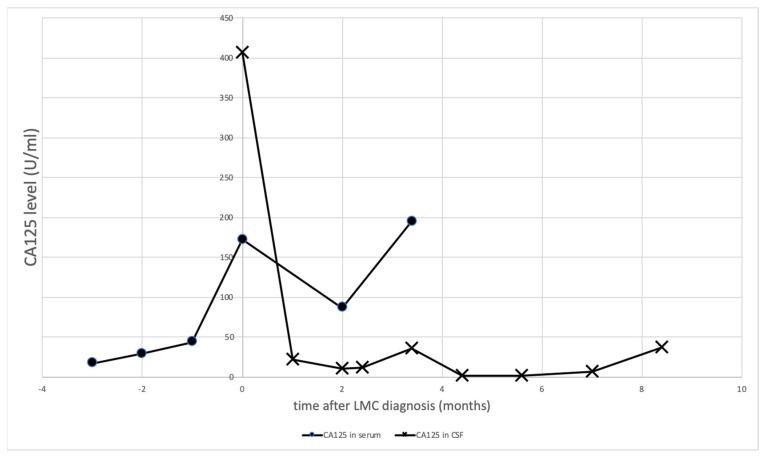
The serum and CSF level of CA125 indicating the controlled CSF lesion by intrathecal administration of MTX.

**Table 1 brainsci-12-00748-t001:** Reported cases of patients with LMC from ovarian cancer.

References	IT * Chmotherapy	Ommaya	Survival after Diagnosis of LM (Months)
Mayer	MTX	(−)	4
Mayer	Thio-tepa	(−)	2
Slosarek	(−)		6
Kaufman	MTX	(−)	<2
Bakri	MTX	(−)	<3
Gordon	MTX	(+)	15
Behnam	(−)		3 weeks
Jackson	(−)		2 days
Stein	(−)		1
Ross	(−)		2 weeks
Plaxe	MTX	(−)	8
Kamiya	MTX	(−)	2
Murphy	(−)		3 days
Khalil	MTX	(−)	15
Delord	MTX	(−)	<1
Ohta	MTX, CDDP	(−)	>13
Chung	(−)		3 weeks
al Barbarawi	(−)		7
Yamanaka	MTX	(+)	>7
Melichar	MTX, AraC	(+)	1
Goto	MTX	(+)	18
Baek	MTX	(−)	8
Eralp	(−)		2 years
Vitaliani	(−)		3 weeks
Hasegawa	(−)		2 weeks
Li	(−)		3 weeks
Toyoshima	(−)		6
Toyoshima	(−)		1
Toyoshima	(−)		8
Krupa	not mentioned		not mentioned
Tahir	(−)		2 days

* IT: intrathecal chemotherapy, LM: leptomeningeal carcinomatosis.

## Data Availability

The data presented in this report are available from the first author upon reasonable request.

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
