# Peer review of "Prolonged Survival and Restored Useful Life by Early Induction of Intrathecal Chemotherapy in a Patient with Leptomeningeal Carcinomatosis from Ovarian Cancer"

_brainsci, 2022, doi:10.3390/brainsci12060748_

Round 1

Reviewer 1 Report

1) Agree that LM from ovarian cancer is an understudied problem, but it is not "extremely rare" in my experience over the past 12 years. I probably see 1-2/year in neuro-oncology as do many of my colleagues across the country. OMit "extremely" from sentence #2 of intro and later. You can imply it was previously felt to be extremely rare. 

2) Did the patient receive concurrent systemic therapy with IT or WBRT or after WBRT? While systemic therapy does not have fantastic penetration, there are definitely regimens that have some CNS penetration(PARP inhibitors, pemetrexed, carboplatin, etc); at 3 months after starting IT therapy (+/- other) was her KPS still 80?

3) I would not conclude that IT therapy was the sole reason for prolonged survival. You did also proceed w WBRT. While data in NSCLC says this does not affect survival, in breast CA, it may indeed affect survival.  Why isnt the improved OS largely from WBRT?

4) Does CA 125 in the CSF match her cytology? her cell count? her protein and glucose? THIS might be the most interesting part of the case report, though need to also discuss other papers looking at this in non-neoplastic patients

5) Would point out that retrospective case series that "support" IT therapy (cited in your discussion) are non-randomized and very biased towards the patients who are doing better, can handle IT therapy, and likely, whose systemic disease is controlled enough that we consider IT therapy. SO, would NOT draw the conclusion that its necessarily the IT chemo without also including the inherent bias in the patients selected for IT therapy. Same goes for the patients selected for an Ommaya v. where you just try a few via LP.

6) Please remove "the text continues here" from the last line of discussion

Author Response

We are very grateful to the reviewers for their insightful comments and suggestions, which would undoubtedly help us to improve our manuscript immensely. As indicated in the responses below, we have taken all their comments and suggestions into account when generating the revised version of the manuscript. Responses to the reviewers’ comments appear after the arrows, in blue text.

Reviewer #1:

1) Agree that LM from ovarian cancer is an understudied problem, but it is not "extremely rare" in my experience over the past 12 years. I probably see 1-2/year in neuro-oncology as do many of my colleagues across the country. OMit "extremely" from sentence #2 of intro and later. You can imply it was previously felt to be extremely rare.

I agree with your opinion. I omitted “extremely” in the manuscript.

2) Did the patient receive concurrent systemic therapy with IT or WBRT or after WBRT? While systemic therapy does not have fantastic penetration, there are definitely regimens that have some CNS penetration(PARP inhibitors, pemetrexed, carboplatin, etc); at 3 months after starting IT therapy (+/- other) was her KPS still 80?

Thank you for your comments.

She did not receive systemic therapy after LMC diagnosis. Her KPS remained at 80 for 3 months after the induction of IT chemo, but the decision of the gynecologist in charge during this period was to not treat systemically, because of the anticipated poor prognosis.

3) I would not conclude that IT therapy was the sole reason for prolonged survival. You did also proceed w WBRT. While data in NSCLC says this does not affect survival, in breast CA, it may indeed affect survival.  Why isnt the improved OS largely from WBRT?

Thank you for your comments.

I agree with your opinion. In this current case, it was the combination of IT chemotherapy and WBRT that prolonged the survival. Whether the elongation of survival owes to the IT chemo, or WBRT, or the synergy of both remains unclear. However, we believe that it was the rapid induction of IT chemo that allowed her a meaningful and relatively symptom free period, and this was largely responsible for this elongated clinical course.

4) Does CA 125 in the CSF match her cytology? her cell count? her protein and glucose? THIS might be the most interesting part of the case report, though need to also discuss other papers looking at this in non-neoplastic patients

Thank you for your comments.

Although we do not have direct evidence whether the patient’s leptomeningeal disease secreted CA125, the serum level of CA125 largely paralleled with the body tumor volume and found it reasonable to suggest that it could be used as a surrogate marker to quantify the amount of leptomeningeal disease. Disease deterioration was accompanied by decrease of glucose, and protein level elevation as usual, but was not quantitatively assessed in this study. According to other papers discussing CSF CA125 level in non-neoplastic patients, the 97.5th percentile was approximately 4.3U/ml, which is reached 4.5 months after LMC diagnosis in our case. I added more detailed explanation about CSF CA125 change in case presentation, and the potential of CSF CA125 as a surrogate marker in discussion (shown in red).

5) Would point out that retrospective case series that "support" IT therapy (cited in your discussion) are non-randomized and very biased towards the patients who are doing better, can handle IT therapy, and likely, whose systemic disease is controlled enough that we consider IT therapy. SO, would NOT draw the conclusion that its necessarily the IT chemo without also including the inherent bias in the patients selected for IT therapy. Same goes for the patients selected for an Ommaya v. where you just try a few via LP.

I agree with the reviewer’s opinion. These case series contain a lot of bias, and the conclusion derived from those cases are also very biased. The indication of IT chemotherapy should have included preserved PS. In this report, we would like to emphasize the potential of IT chemotherapy via Ommaya reservoir for restoring independent and useful life in LMC patient, who mostly live miserable remaining life without treatment for LMC even with preserved PS at diagnosis. For clarity, I added sentence to explain those in discussion part (shown in red).

6) Please remove "the text continues here" from the last line of discussion

Thank you very much for finding my mistake. I omitted the sentence.

Reviewer 2 Report

Well written case report on meningeosis carcinomatosa in ovarian cancer, which is very rare in ovarian cancer (32 cases reported) therefore it is worth to publish. 

Author Response

We are very grateful to the reviewers for their insightful comments and suggestions, which would undoubtedly help us to improve our manuscript immensely. As indicated in the responses below, we have taken all their comments and suggestions into account when generating the revised version of the manuscript. Responses to the reviewers’ comments appear after the arrows, in blue text.

Reviewer #2:

Well written case report on meningeosis carcinomatosa in ovarian cancer, which is very rare in ovarian cancer (32 cases reported) therefore it is worth to publish.

Thank you very much for your review.
